# Augustine's *Enchiridion*: An Anti-Pelagian Interpretation of the Creed

David Burkhart Janssen 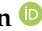

Evangelisch-Theologische Fakultät, Universität Tübingen, Liebermeisterstr. 12, 72076 Tübingen, Germany; david-burkhart.janssen@uni-tuebingen.de

**Abstract:** At first glance, Augustine did not combine his soteriology and his Trinitarian doctrine in his anti-Pelagian oeuvre. Therefore, this article pursues the more hidden and implicit connections between these topics. The starting point of this endeavour is an analysis of the *Enchiridion*, a catechetical work in which Augustine interpreted the Roman—later so-called Apostle's—Creed. Simultaneously, Augustine directed his attention in the *Enchiridion* to questions and arguments which originate from the Pelagian controversy such as original sin, grace, baptism, remission of sin(s) and the theory of predestination. Thus, this article ponders the question of how Augustine reflected his Trinitarian doctrine within this anti-Pelagian soteriology. While Augustine seldom referred to his Trinitarian doctrine explicitly in the *Enchiridion* (and his anti-Pelagian oeuvre), he presented in these works a conception of how the triune God operates as creator and saviour. This anti-Pelagian concept of God seizes several aspects which also appear in Augustine's *De trinitate*. Moreover, by emphasising the unity of God's operation as creator and saviour against the Pelagians, Augustine argued in favour of a specific Trinitarian doctrine: *opera trinitatis ad extra inseparabilia*. Thus, this article finally tries to analyse how Augustine amalgamated his anti-Pelagian Christocentric soteriology with his Trinitarian doctrine.

**Keywords:** Augustine; Enchiridion; Pelagianism; Trinitarian doctrine; Christology; soteriology/doctrine of grace





## 1. Introduction

In his anti-Pelagian oeuvre, Augustine seldom referred to Trinitarian terminology and doctrine explicitly. Instead, he mainly constructed Pelagianism on the basis of anthropological, soteriological and Christological topics. This is due to the fact that both Augustine and his "Pelagian" opponents accepted and promoted the Trinitarian faith of Nicaea and Constantinople. While questioned at the synod of Diospolis, Pelagius explicitly referred to his orthodox Trinitarian faith (Pelagius in Aug. *gest. Pel.* 43/99, 1–2: *Ego enim in unius substantiae trinitatem credo et omnia secundum doctrinam sanctae catholicae ecclesiae*). Pelagius pursued the same strategy in his *Libellus fidei* 1–8 (377, 1–378, 18). Augustine was aware of Pelagius's and Caelestius's Trinitarian confession (*gr. et pecc. or.* 1, 35/152, 18–23; 2, 26/185, 1–22). Already before the Pelagian controversy, the British monk had written a work called *De fide trinitatis* arguing against the "Arian" heresy (cf. Dupont 2008). Julian of Aeclanum also employed a Trinitarian confession as an argument for his orthodoxy. According to Julian, the triune God is just by definition. Those who teach a natural sin—in Julian's opinion, the Manichaeans and Augustine—consequently reject the just creator and, therefore, the Trinitarian faith (Julian, *Ad Florum* in Aug. *c. Iul. imp.* 1, 28–32.75/23, 1–24, 5; 90, 1–91, 35). Augustine defended himself against this charge by distinguishing between good creation and postlapsarian sinful state (Aug. *c. Iul. imp.* 1, 28–32/23, 1–24, 11; cf. *ench.* 3, 9-4, 15/52, 1–56, 94). Thus, both Julian and Augustine did not address a problem of the Trinitarian doctrine but the problem of theodicy.

Augustine, however, connected theological ideas from different controversies and discussions. While he did not explain his Trinitarian doctrine within his anti-Pelagian

works, he employed, for example, the soteriological arguments which he developed within his anti-Pelagian oeuvre in *De trinitate* (cf. Drecoll 2000; Yam 2019; Janssen forthcoming, chp. 8.4.1).

Another work in which Augustine amalgamated his anti-Pelagian argumentation with his Trinitarian doctrine is, as I will demonstrate, the *Enchiridion* (cf. Section 2). By focussing on the *Enchiridion*, I will analyse how Augustine combined his anti-Pelagian soteriology[1] and his Trinitarian doctrine (cf. Section 3). As a result, I will raise the question for which reasons Augustine began to combine his anti-Pelagian soteriology explicitly with his Trinitarian doctrine in the *Enchiridion* and other works written in the 420 s (cf. Section 4).

## 2. The Double Structure and the Anti-Pelagian Character of the *Enchiridion*

Augustine wrote the *Enchiridion* around 421, in the midst of the Pelagian controversy (TeSelle 1996–2002, p. 1324). This work, which is also called *De fide, spe et caritate* (Aug. *retr.* 2, 63/140, 1–3), is a masterpiece, demonstrating Augustine's particular ability to communicate his theological thoughts to a wider audience. Therefore, it was received and interpreted as a summa of Augustine's whole theology.[2] This characterisation, however, does not match Augustine's own approach and concept.[3] I will argue, instead, that the *Enchiridion* is especially based on Augustine's anti-Pelagian argumentation. Although Augustine did not refer in the *Enchiridion* explicitly to the Pelagian controversy, the *Enchiridion* can be numbered to his anti-Pelagian oeuvre in a broader sense.

As is the case with many of his works, Augustine wrote the *Enchiridion* to comply with a specific request: Laurentius, the brother of the tribune Dulcitius (Aug. *Dulc. qu.* 1, 10/265, 261–262; cf. *ench.* 1, 1/49, 1), thus a member of the Roman elite, had requested a handbook (*enchiridion*) containing all the necessary facts of the orthodox faith.

> Aug. *ench.* 1, 4 (49, 29–50, 36): *Uis enim tibi, ut scribis, librum a me fieri quem Enchiridion, ut dicunt, habeas et de tuis manibus non recedat, continens postulata, id est quid sequendum maxime, quid propter diuersas principaliter haereses sit fugiendum, in quantum ratio pro religione contendat, uel quid in ratione cum fides sit sola non ueniat, quid primum quid ultimum teneatur, quae totius definitionis summa sit, quod certum propriumque fidei catholicae fundamentum.*

Augustine began his answer by emphasising that true wisdom (*sapientia*) is the worship of God (θεοσέβεια/*cultus Dei*) (Iob 28:28) (Aug. *ench.* 1, 2/49, 9–20).[4] He identified θεοσέβεια with the triad from 1Cor 13: *fides, spes, caritas* (Aug. *ench.* 1, 3/49, 21–28).[5] These three are interwoven with each other; however, according to Augustine, it is required to answer what to believe in (*quid credendum/credi*) before one could discuss what to love (*quid amandum/amari*) and what to hope for (*quid sperandum/sperari*) (cf. Aug. *ench.* 1, 4-2, 8/50, 36–52, 62). Christians hope and love due to their faith (*quod speramus et amamus credendo uenturum esse* [*ench.* 2, 8/52, 58]). Therefore, hope and love cannot exist without faith (*proinde nec amor sine spe est nec sine amore spes, nec utrumque sine fide* [*ench.* 2, 8/52, 61–62]). However, faith without hope and love is null and void (Jak 2:19) (cf. Aug. *f. et op.* 23/64, 6–18).

By equating *sapientia* and θεοσέβεια, Augustine connected knowledge and faith (Yam 2019, p. 473) which operates through love (*fide[s] quae per dilectionem operatur* [Aug. *ench.* 1, 5/50, 48]). Already in *De trinitate*, "knowledge and love are two key elements of Augustine's Trinitarian argument" (Yam 2019, p. 342; cf. Kany 2007, pp. 181–90). Augustine conceptualised the *Enchiridion* as a work containing those beliefs which are required to revere God ("*fides quae*"). However, he confined this benefit of the *Enchiridion* as well as of other catechetical writings by emphasising the necessity of a loving faith in one's own heart (cf. Harmless 1995, pp. 368–75).

### 2.1. The Trinitarian Structure

The bishop of Hippo arranged the *Enchiridion* mainly based on the Roman, respectively, Milanese Creed (Aug. *ench.* 2, 7/51, 1.9–10) (cf. Kinzig 2013, p. 104; Kinzig 2017, pp. 300–10;

Westra 2002, pp. 189–96), thus using a Trinitarian structure (Kinzig 2013, pp. 124–26). Augustine began with a short introduction of the trinity and, thereby, introduced God as good creator and saviour (Aug. *ench*. 3, 9–10, 33). Afterwards, he transitioned to salvation achieved by Christ's incarnation and death (Aug. *ench*. 10, 33-14, 55). However, Augustine mentioned the final aspects of the second part of the Creed (Christ's second coming) only briefly (Aug. *ench*. 14, 54–55). Thirdly, and finally, he referred to the Holy Spirit (Aug. *ench*. 15, 56). In this context, Augustine elaborated on ecclesiology (Aug. *ench*. 15, 57-16, 63), the remission of sins (Aug. *ench*. 17, 64-22, 83), the resurrection of the flesh (Aug. *ench*. 23, 84-29, 110) and eternal life (Aug. *ench*. 29, 111–113). At first glance, this structure has recourse to Trinitarian appropriations which are inherent to the Creed itself: the Father as creator, the Son as saviour, the Holy Spirit as inspirer and consummator.

### 2.2. The Soteriological Structure

By emphasising θεοσέβεια (*cultus Dei*) as his main theme, Augustine gave his attention to soteriology. Therefore, he structured his argumentation in the *Enchiridion* according to his *"Heilsstadienlehre"* (anthropological–soteriological stages) (Drecoll 1999, pp. 147–211, 356–58; Janssen forthcoming, chp. 2.6; cf. also, Rivière 1942, pp. 107–10). He began with the creation and its prelapsarian good condition (*ante peccatum*), thereby arguing that evil is nothing but a *priuatio boni* (Aug. *ench*. 3, 9-4, 1 5). As an intermediate argument, Augustine inserted an excursus on epistemology, how to distinguish good and evil (*causae cognoscendae sunt rerum bonarum et malarum* [Aug. *ench*. 8, 23/63, 2–3]), as well as truth and falsehood.[6] Without this differentiation, one cannot state and defend orthodox assertions (*at si tollatur assensio fides tollitur, quia sine assensione nihil creditur* [Aug. *ench*. 7, 20/61, 28–29]) (Aug. *ench*. 5, 16-8, 23). With this excursus, Augustine referred to the connection of knowledge (*notitia*) and faith (*fides*) (Aug. *ench*. 6, 18/59, 34–35). Afterwards, Augustine explained the fall of the angels and of Adam which results in the postlapsarian condition of humanity (*peccatum originale*) (Aug. *ench*. 8, 24-9, 29).

According to Augustine, postlapsarian humans are not able to change their evil will to good (Aug. *ench*. 9, 30); thus, it belongs to God's grace alone to redeem and save (Aug. *ench*. 9, 30-10, 33). This gracious act, i.e., the change from the stage *sub peccato* to *sub gratia*, is achieved by Christ as mediator and reconciler (Aug. *ench*. 10, 33-13, 41). This argumentation resembles major aspects of Augustine's anti-Pelagian one, mostly referring to the same biblical quotations.[7] By baptism, man *sub peccato* participates in Christ's incarnation and death (Aug. *ench*. 13, 42-14, 55). Thereby, Augustine employed the Adam–Christ–typology (Rom 5:16.18) (Aug. *ench*. 14, 50–51) which is fundamental to his anti-Pelagian reasoning scheme (Aug. *pecc. mer*. 1, 9–21/10, 8–21, 23; *c. ep. Pel*. 4, 7–8/527, 8–529, 28; cf. Lamberigts 2005, pp. 174, 193).

After presenting his Christology, the bishop of Hippo focussed on the state *sub gratia*: through Christ's death, grace and baptism, humans become part of the *ciuitas Dei* which exists both celestially and terrestrially (Aug. *ench*. 15, 56-16, 63). Augustine especially emphasised that the remission of sins is still necessary for Christians (Aug. *ench*. 17, 64-22, 83). Thereby, Augustine seized several biblical arguments against the possibility to live without sin from his anti-Pelagian oeuvre (cf. Dupont 2003, especially, pp. 364–413, 592–603; Janssen forthcoming, chp. 7):

- 1Joh 1:8 (Aug. *ench*. 17, 64/84, 16–25; *c. ep. Pel*. 3, 4/488, 21–489, 20);
- Mt 6:12 (Aug. *ench*. 19, 71–74/88, 9–89, 75; *c. ep. Pel*. 1, 27–28/445, 24–447, 6; 3, 5/490, 11);
- Mt 6:13 (Aug. *ench*. 22, 81/94, 1–95, 20; *c. ep. Pel*. 1, 27/446, 15–17);
- 1Cor 13:12 (Aug. *ench*. 16, 63/83, 47–62; *c. ep. Pel*. 3, 17/506, 12–16; 3, 21/511, 26–28).

The final chapters (Aug. *ench*. 23, 84-27, 107) examine the change between the stage *sub gratia* and the eschatological stage *in pace* (resurrection of the flesh, final judgement and eternal life). Interestingly, Augustine inserted his theory of predestination at this point and not—as could be suggested—at the passage which regards the change from *sub peccato* to *sub gratia*. Augustine underlined that the entry into life eternal completely depends on

God's mercy. Furthermore, Augustine argued epistemologically: God's decision whom he had predestinated and whom not will only be revealed in the eschaton. Thereby, he also revealed a pastoral interest: Christians should trust in God's mercy and not in themselves (cf. Hombert 1996, pp. 324–39; van Geest 2011, pp. 193–210).

Although Augustine chose the Creed as the main structural element of the *Enchiridion*, he employed his *"Heilsstadienlehre"* for his argumentative structure: *ante peccatum—sub peccato—sub gratia—in pace*, thus, a soteriological scheme (cf. Aug. *ench.* 31, 118/112, 42–44: *Harum quattuor differentiarum prima est ante legem, secunda sub lege, tertia sub gratia, quarta in pace plena atque perfecta*).[8] This soteriological structure shows a high reliance on anti-Pelagian argumentation. Thus, Augustine amalgamated in the *Enchiridion,* a Trinitarian and a soteriological perspective. As a result, the *Enchiridion* significantly differs from other Augustinian interpretations of the Creed such as *De fide et symbolo* (393),[9] *De agone christiano* (396) or *De symbolo ad catechumenos* (after 415).[10]

### 3. Trinitarian Argumentation in the *Enchiridion*

Although Augustine introduced the Trinitarian Creed as the main structure of the *Enchiridion*, the line of argumentation primarily follows his soteriological doctrine. Thus, the *Enchiridion* is more anti-Pelagian than it might seem at first glance. Moreover, Augustine employed his Trinitarian doctrine in the *Enchiridion* in quite similar fashion than in his anti-Pelagian oeuvre. In the following, I will expose four aspects of how Augustine presented his Trinitarian doctrine in the *Enchiridion* as well as in his anti-Pelagian corpus:

1.  Although the *Enchiridion* is an interpretation of the Creed, Augustine near-completely omitted explicit Trinitarian terminology. Thereby, the *Enchiridion* closely resembles the anti-Pelagian works and significantly varies from other Augustinian interpretations of the Creed.

2.  Instead, Augustine focussed on the exceptional role of Christ as saviour and foundation of faith. According to Augustine, Christ's sacrifice is the only means of salvation. Moreover, Augustine referred to the Holy Spirit as *donum Dei/caritatis* which transforms humans by grace. In contrast thereto, Augustine did not attribute any particular role to God the Father (*pater*) alone.

3.  Simultaneously, Augustine argued that God's operation *ad extra* is inseparable. He stated that the triune God is likewise creator and saviour, especially, by attributing creation, salvific will, predestination and just judgement to God (*Deus*) without distinguishing the persons of the trinity.

4.  Although Augustine structured the *Enchiridion* based on the Trinitarian Creed, he emphasised Christology as the centre piece of the salvific faith. This corresponds with the fact that he composed the *Enchiridion* as a catechetical work. Thereby, the particular role of Christ is interrelated with a Christ-centred piety: Salvation is participation in Christ. However, Augustine especially referred to his Trinitarian doctrine with regard to his Christology. While presenting Christ's incarnation and death as the core of salvation, Augustine interpreted the reconciliation achieved by Christ as an act of the whole trinity.

As shall be shown in Section 4, this approach in the *Enchiridion* is not an isolated case. During the later Pelagian controversy, Augustine increasingly embedded his Christology with his Trinitarian doctrine.

### 3.1. Explicit Trinitarian Terminology in the Enchiridion

Augustine seldom referred to the trinity (*trinitas*) in the *Enchiridion* explicitly. More often, he used the term *Deus*. He identified the trinity as the *Deus unus et uerus* (Aug. *ench.* 3, 9/53, 17; cf. Ayres 2010, p. 103). In particular, he employed the term *Deus* while referring to God as creator (Aug. *ench.* 3, 9–11), as just judge and as distributor of grace who predestinates those (Aug. *ench.* 24, 95-29, 112) who will dwell in his eschatological kingdom (*regnum Dei*) (Aug. *ench.* 18, 67-19, 70/86, 28–87, 8).

In *Contra duas epistulas Pelagianorum,* Augustine proceeded in a similar way. In book 2 of this work, Augustine depicted his theory of predestination in more detail than in the precedent anti-Pelagian works. Thereby, he mostly attributed the predestination to God's inscrutable will (cf. Aug. *c. ep. Pel*. 2, 12/472, 21–473, 8). Only in the final chapters did Augustine briefly introduce a Christological interpretation of God's predestination (Aug. *c. ep. Pel*. 2, 22/484, 12–22) (cf. Sections 3.4 and 4).

When Augustine addressed his Trinitarian doctrine in the *Enchiridion,* he did not employ Trinitarian triads or a specific Trinitarian terminology (*substantia*, *essentia*, *persona* et al.) (cf. Drecoll 2007, pp. 450–56; Kany 2007, pp. 198–210, 227–40). Instead, these terms only appear in the *Enchiridion* referring to the Christological union. Only in some central passages did Augustine explicitly dwell on his Trinitarian doctrine. In *ench*. 3, 9–10, he identified the belief in the triune God as creator of all heavenly and earthly things as the content of faith (*quid credendum sit quod ad religionem pertineat* [Aug. *ench*. 3, 9/52, 1–2]). Thereby, Augustine briefly referred to the intra-Trinitarian relations: The Son is begotten (*gentius*) and the Spirit proceeds (*procedens*) from the Father but is the Spirit of both Father and Son (Aug. *ench*. 3, 10/53, 18–21; 12, 38/70, 2–71, 3.12–13; cf. Kany 2007, pp. 198–210, 216–27). In most of his interpretations of the Creed, Augustine laid the focus on intra-Trinitarian relations far more extensively (cf. Aug. *symb. cat*. 3–5/186, 52–189, 31).

While presenting his Christology, Augustine focussed on a central idea of his Trinitarian doctrine: *Neque enim separabilia sunt opera trinitatis* (Aug. *ench*. 12, 38/71, 21). Consequently, Augustine stated that the whole trinity is active within the process of salvation (Aug. *ench*. 15, 56/79, 7–80, 42). This also has ecclesiological consequences: the *ciuitas Dei* is bound together through the *uinculum caritatis*. Thus, the Church is one *consortio aeternitatis* because the whole trinity operates in the Church (Aug. *ench*. 15, 56/79, 9–20; cf. Ployd 2015, pp. 187–88, who analyses this argument in Augustine's anti-Donatist works).

The *Enchiridion* does not contribute new ideas to Augustine's Trinitarian doctrine. Instead, Augustine adopted lines of argumentation from *De trinitate*[11] as well as from his anti-Pelagian works and remodelled them by combining soteriological and Christological arguments with a Trinitarian structure. Thereby, Augustine used two different approaches of Trinitarian doctrine simultaneously: firstly, he attributed specific operations to one person of the trinity (cf. Section 3.2), secondly, he emphasised that the operations of the triune God *ad extra* are indivisible (cf. Section 3.3).

*3.2. Trinitarian "Appropriations"*

Although Augustine did not extensively outline his Trinitarian doctrine in the *Enchiridion,* he mentioned the three persons of the trinity frequently: *pater*, *filius* and *spiritus sanctus*. Thereby, he combined *Deus* and *pater* (Aug. *ench*. 10, 34/68, 30; 12, 38–39/70, 2–71, 6.12–13; 72, 43)[12] and *Deus* and *Christus* (Aug. *ench*. 12, 38/71, 7–16), especially while discussing Christology. In this context, he also used the term *Deus* to signify that Christ, the mediator between God and humans (cf. Remy 1979), reconciled humanity with God (*reconciliari Deo* [Aug. *ench*. 10, 33/68, 15–25]; cf. Section 3.4). However, Augustine rarely used the term *pater* besides explicit Trinitarian and Christological passages. Due to the Trinitarian statement in *ench*. 3, 9–10, it seems improbable that Augustine used the term *Deus* to refer exclusively to the Father (*pater*) (cf. Madec 1996–2002, pp. 317–19).[13] In the *Enchiridion,* Augustine did neither appropriate the impulse of the creational act to God the Father nor did he characterise God the Son (*Uerbum*)[14] as the mediator of creation (cf. Aug. *ciu*. 11, 24/343, 1–344, 43).

While Augustine employed the term *filius (Dei)* within a Christological context only (Aug. *ench*. 9, 30; 10, 33–13, 41; 21, 80), he referred to the exceptional role of Christ as saviour throughout the *Enchiridion*: Augustine called Christ the fundament of faith (1Cor 3:11) (Aug. *ench*. 2, 5/50, 54–62). He therefore could attribute the salvific faith to Christ (*uiam* . . . *qua imus ad Deum quae uia fides est Christi quae per dilectionem operatur*) (Aug. *ench*. 7, 21/61, 48–49; cf. 18, 67/86, 29–30: *propter fidem Christi salui erunt*) (cf. Studer 1993, pp. 87–102). According to Augustine, Christ operates knowledge and love likewise. For this reason,

Augustine rejected a strict notion of salvation by faith, i.e., a doctrine of salvation by faith without works, as heretic interpretation of 1Cor 3 (Aug. *ench.* 18, 67–68/85, 1–87, 73; cf. *f. et op.* 27/69, 1–72, 3).

Moreover, Augustine mainly interpreted soteriology Christologically (*ench.* [9, 30-]10, 33-14, 53),[15] especially, by emphasising Christ's atoning death: Augustine framed his discussion of Christ's person (*ench.* 10, 34-12, 40) with references to his crucifixion (*ench.* 10, 33/68, 11–25 and 13, 41/72, 1–27). Thereby, Augustine entangled Christ's person as *mediator* and as God–human with Christ's work as reconciler (*reconciliator*). According to Augustine, baptism (Aug. *ench.* 14, 52–53/76, 46–78, 111) and incorporation in the Church (Aug. *ench.* 16, 61–62/82, 21–46) rely on the salvific blood of Christ. Whoever is rescued in the Last Judgement will be saved by Christ's gracious death (Aug. *ench.* 28, 108/107, 66–108, 83). Especially in his anti-Pelagian oeuvre, Augustine emphasised Christ's death as a unique means of salvation (cf. Janssen 2023).[16]

In the *Enchiridion*, Augustine often mentioned the Holy Spirit combined with Christ. He especially introduced the Spirit as the driving force within Christ's incarnation (Aug. *ench.* 11, 37-12, 40/70, 31–72, 64). Moreover, he attributed the transformation in baptism to the Holy Spirit (Aug. *ench.* 14, 49/76, 11–14; 17, 64–65/83, 15–84, 41; 19, 71/88, 12; 20, 75/90, 15–20; cf. Lamberigts 2009) whom he called the gift of love and God's gift (*donum Dei*) (Aug. *ench.* 11, 37/70, 33; 12, 40/72, 62–63). Thereby, Augustine referred to the Pneumatological aspects of the soteriology which he even further developed in his anti-Pelagian works, especially *De spiritu et littera* (cf. Yam 2019, pp. 447–450, 492–493, 602–616; Kantzer Komline 2020, pp. 331–51), *Contra duas epistulas Pelagianorum* (cf. Aug. *c. ep. Pel.* 3, 11–12/497, 10–499, 27) and in *De trinitate* (cf. Aug. *trin.* 15, 31–32/505, 96–508, 32).

Augustine employed Trinitarian "appropriations" mainly while referring to soteriology. He especially highlighted the role of Christ's salvific death in which Christians participate. While discussing the creation and God's will (predestination, judgement), Augustine did not accentuate one of the trinity but referred to the triune God.

### 3.3. Triune God as Creator and Saviour

One of Augustine's central arguments in the *Enchiridion* is that God (*Deus*) is likewise creator and saviour. Following the first part of the Creed (*credo in Deum, patrem omnipotentem*), Augustine stated that the triune God (*trinitas, Deus*) has created all things visible and invisible (Aug. *ench.* 3, 9–11). Although all humans have deserted their creator through sin (Aug. *ench.* 8, 27), God, the creator (*placuit itaque uniuersitatis creatori atque moderatori Deo* [Aug. *ench.* 9, 29/65, 13–14]), shows his mercy upon humans. This might be the reason why Augustine introduced Christ as redeemer (Aug. *ench.* 10, 33/68, 13–25) only after he had already explained his theory of *gratia gratuita*. Although only the second person of the trinity, the Son incarnated and died to reconcile fallen humanity with God, the whole trinity is involved in Christ's incarnation and salvific death (Aug. *ench.* 12, 38). Within Christ's incarnation and death, the trinity is revealed. Consequently, Augustine emphasised that neither Christ nor the Holy Spirit alone dwells in Christians but the trinity (Aug. *ench.* 15, 56/79, 4–80, 42; cf. *Ep.* 187, 16–21/93, 18–100, 5; *trin.* 7, 6/254, 85–109; 15, 32/507, 1–508, 32).

In the *Enchiridion*, Augustine stressed the unity of God's operation *ad extra* as being more pronounced than the unity of God's being.[17] Therefore, Augustine's main opponents in the *Enchiridion* cannot be identified with the Homoians ("Arians") or other "antitrinitarian" heresies although Augustine began to engage an Homoian theology, in 419, directly (cf. Sieben 2008, pp. 9–30).[18] Augustine only defined and defended his doctrine of God by introducing core elements of his Trinitarian doctrine.

Augustine's argumentation of emphasising the unity of God's operation is instead directed against Manichaeism and Pelagianism (cf. Janssen forthcoming, chp. 2.6). On the one hand, Augustine rejected the Manichaean dualism by stating that the one and good God is not only the saviour, but also the creator. For distinguishing creator and creation, Augustine referred already in *Contra Faustum* to the inseparability of the trinity, however,

regarding the being of God, and not God's operations (cf. Aug. *c. Faust*. 20, 7/541, 4–542, 16). Due to the difference between creator and creation, the latter could be perverted (Aug. *ench*. 4, 12/54, 1–26). Thus, sin is nothing but a *priuatio boni* resulting from the angels'[19] and from Adam's voluntary transgression of God's will. On the other hand, Augustine seized an anti-Pelagian argument: the creator of all is as well the saviour because all humans require grace to be redeemed (cf. Aug. *nat. et gr*. 39/261, 28–262, 2). God's operation of salvation is not inherent in God's creational acts (i.e., the created abilities of human nature) but surpasses creation (cf. Aug. *gr. et pecc. or*. 1, 3–10/126, 23–133, 23). While the Manichaeans would introduce a heretical dualism rejecting God as creator, the Pelagians would deny God as saviour by rejecting the postlapsarian state of humanity.

> Aug. *c. ep. Pel*. 2, 2, 2 (461, 11–16): *Manichei dicunt Deum bonum non omnium naturarum esse creatorem, Pelagiani dicunt Deum non esse omnium aetatum in hominibus mundatorem, saluatorem, liberatorem. Catholica* utrosque *redarguit et contra Manicheos defendens Dei creaturam, ne ab illo instituta negetur ulla natura, et contra Pelagianos, ut in omnibus aetatibus perdita requiratur humana natura.*

Moreover, Augustine demonstrated the unity of God's operation by stating that God (*Deus*) is likewise the distributor of grace and the just judge (Aug. *ench*. 24, 94–29, 113). Thereby, Augustine defended himself against complaints that his doctrine of predestination is propagating a concept of God as unjust tyrant. These charges against the theory of predestination became public knowledge during the Pelagian controversy, especially after 418 (Julian of Aeclanum, *Ep. ad Rufum fr*. 4//337, 28–32 and 22/339, 98–102 [in *c. ep. Pel*. 2, 10/469, 13–21]; cf. Janssen forthcoming, chp. 3.5).

*3.4. Trinitarian Christology (ench. 10, 33–13, 42)*

All three aspects which I very shortly summarised in the Sections 3.1–3.3 show a striking resemblance between the *Enchiridion* and Augustine's anti-Pelagian argumentation: Augustine mostly omitted his Trinitarian terminology. He focussed on Christ and in some cases on the Holy Spirit to present his soteriology. While speaking about creation and predestination, Augustine employed the term *Deus* without distinguishing the persons of the trinity. In general, Augustine especially focussed on the second aspect while refuting and attacking the Pelagian argumentation.

In the *Enchiridion,* Augustine argued that due to the oneness of God (*Deus*), God also operates as unity and is likewise creator and saviour, judge and redeemer. Therefore, it might seem inconsistent that according to Augustine, Christ operates as the unique saviour. Ensuing from this Trinitarian problem, Augustine raised two questions: How can the second person of the trinity be the only one of the three members of the "immanent trinity" who became incarnate when the trinity operates *ad extra* inseparably ("oecumenic trinity") (cf. already Aug. *trin*. 1, 7–8/34, 1–37, 23)? How can the two natures in Christ exist and operate together when God alone obtains grace? These problems are both related because, according to Augustine, grace is a divine operation of the trinity, neither resulting from one person of the trinity alone nor from human merits.

In this context, Augustine referred to the Creed: *ut crederemus in Dei patris omnipotentis unicum filium, natum de spiritu sancto et uirgine Maria* (Aug. *ench*. 10, 34/68, 29–30; cf. 12, 38/71, 19–24). Augustine stated that the divinity of Christ was neither changed nor diminished by the incarnation (Aug. *ench*. 10, 34/68, 31–32); nor did Christ accept humanity only partially. In this short comment, Augustine rejected both Arianism and Apollinarianism (cf. Aug. *c. s. Arrian*. 5/191, 11–25; cf. Brennecke 2008, 2015). Due to the birth of the virgin's womb, Christ was, however, without concupiscence and sin (Aug. *ench*. 10, 34/68, 32–69, 47). By referring to Phil 2, Augustine emphasised the unity within Christ (*unitas personae; ex utroque unus Christus*) (Aug. *ench*. 10, 35/69, 48–65) (cf. Verwilghen 1985; Kantzer Komline 2022). Therefore, Augustine distinguished between God's being (*natura*) and God's creational and salvific operation (*gratia*). While incarnating Christ remained (*manere*) divine (*forma Dei*) what he is by nature (*natura*). He deigned to become human (*forma serui*) due to grace (*gratia*). Therefore, Christ's human nature is a product of grace,

not of merit (Aug. *ench*. 11, 36/70, 13–30). In this sense, Christ's human nature is a new creation (*factus est et hominis filius* [Aug. *ench*. 10, 35/69, 55; cf. *Ep*. 137, 9–10/107, 14–109, 14]), however, not *ex nihilo*. God's creational grace is operating together with the virgin Mary (Lc 1:28–30) (Aug. *ench*. 11, 36/70, 13–30; 11, 38/71, 30–35). Therefore, Christ's human nature is without the postlapsarian defect and, instead, full of grace (*plenus gratia*) (Aug. *ench*. 10, 35/69, 55; 11, 36/70, 27).

Already, in *De peccatorum meritis*, Augustine described Christ's incarnation as an act of grace (Aug. *pecc. mer*. 1, 60–62/59, 24–63, 25). With Christ's incarnation as an example, Augustine tried to convince his audience that God distributes grace graciously and without responding to human merits. However, Augustine did not consider, at this point, how the trinity operates in Christ's incarnation. In *De peccatorum meritis* 1, 60–62, Augustine mainly laid his focus on his Christocentric soteriology and not on Christology or on the understanding of God. After *De peccatorum meritis*, Augustine abandoned this argument in his anti-Pelagian argumentation. By referring to Lc 1:28.30 in the *Enchiridion*, Augustine even extended this soteriological interpretation of Christ's incarnation. He reused this argumentation in several of his later writings to characterise the relationship between Christ and the Holy Spirit (cf. Aug. *praed. sanct*. 1, 30–31/205, 1–207, 19; *trin*. 15, 46/526, 45–527, 75; *Io. eu. tr*. 74, 1–3/512, 19–514, 19) (cf. Section 4). Following the argument, that Christ's human nature is full of grace, Augustine even stated that Christ's human nature is predestinated (cf. Aug. *praed. sanct*. 1, 30/205, 1–2: *Est etiam praeclarissimum lumen praedestinationis et gratiae, ipse saluator, ipse mediator Dei et hominum homo Christus Iesus*; cf. *praed. sanct*. 1, 31/206, 8–9; 207, 19–20 and 2, 67/270, 1–3). Thus, Augustine emphasised in the *Enchiridion* that within the act of incarnation, God not only operates as creator and saviour but affiliates himself, i.e., his nature, with the creation in the act of salvation.

Augustine did not regard the two natures of Christ as something merely theoretical or abstract. He mostly avoided speaking of two natures or substances (cf. Aug. *ench*. 11, 38/71, 11–12: *utraque substantia, diuina scilicet atque humana*) and emphatically rejected the idea of two sons (*duo filii*) (Aug. *ench*. 10, 35/69, 63–64) (cf. van Bavel 1954, p. 45; Drobner 1986, p. 250).[20] According to Augustine, Christ's unity is a union of two properties (*aliud . . . aliud* [Aug. *ench*. 10, 35/69, 60–62]) and two identities (*forma[] Dei . . . forma[] serui*; *filius Dei . . . filius hominis* [Aug. *ench*. 10, 35/69, 58–59.62–63]), i.e., also two wills (dyotheletism [cf. Aug. *c. s. Arrian*. 6/192, 16–199, 58; cf. Berthold 2013; Kantzer Komline 2020, pp. 282–98]). The unchangeable divine identity is the Son who humbles himself. The human identity (*filius serui/hominis*) is an individual human who was newly created through God's grace when he was assumed by God the Son through the virgin Mary. Therefore, this human identity of Christ is in the same prelapsarian state as Adam (without concupiscence and sin) but persistently full of grace. Thus, Christ's condition surpasses Adam's paradisal one. Due to grace (and not to merits), Christ's human identity agrees with the divine.[21] Thus, the two identities of Christ are united (*unus hominis filius, idemque Dei filius* [Aug. *ench*. 10, 35/69, 62–63]) because they are one (and not two):[22] Augustine claimed that assertations about Christ's human nature are likewise assertations about the one person of the incarnated Christ (*unitas personae unigeniti* [cf. Drobner 1986, pp. 241–74]) (Aug. *ench*. 15, 56/79, 9–10). The one who is grace by nature received the human nature by grace (Aug. *ench*. 11, 36/70, 24–30).

Furthermore, Augustine argued that the formulation *natum de spiritu sancto* cannot be interpreted to mean that Christ was born as human from the Holy Spirit in the same way that God the Son is begotten from God the Father (Aug. *ench*. 12, 38/70, 1–71, 9). Christ's human nature received grace by the triune God because the triune God is the creator and saviour. On the one hand, Augustine explained the particular mentioning of the Holy Spirit in the Creed by referring to Lc 1:35 and Mt 1:20. On the other hand, he stated that the emphasis on the Holy Spirit represents God's distribution of grace:

> *Quae gratia propterea per spiritum sanctum fuerat significanda quia ipse proprie sic est Deus ut dicatur etiam Dei donum; unde sufficienter loqui, si tamen id fieri potest, ualde prolixae disputationis est.* (Aug. *ench*. 13, 40/72, 61–64; cf. 11, 37/70, 35–37)

According to Augustine, Christ is both the example and exceptional in relation to humanity and divinity. On the one hand, Christ's human nature is the prime example for the operation of the *gratia gratuita* without merits (cf. Geerlings 1978, pp. 211–28; McWilliam Dewart 1982; Kantzer Komline 2020, pp. 299–308). Augustine even stated that by the example of Christ, the character of grace becomes evident (Aug. *ench*. 11, 36/69, 1–2).[23] On the other hand, Christ's human nature is exceptional because—due to a divine act of grace (Aug. *ench*. 12, 40/72, 59–64)—only Christ is conceived without concupiscence and sin (*similitudo carnis peccati* [Rom 8:3]) (Aug. *ench*. 10, 34/68, 35–69, 45; 13, 41/72, 1–73, 27; cf. *praed. sanct*. 1, 31/206, 13–207, 19). Augustine differentiated between the unique incarnation and God's gracious acts by which sinners are reborn in the state of grace (cf. Aug. *Ep*. 187, 10/89, 6–90, 2; 187, 39–40/116, 9–118, 2). Thus, Augustine emphasised that no Christian will ever be able to become like Christ because Christ is the only God–man and the only sinless human. Instead, as he especially stated in his anti-Pelagian oeuvre, humans permanently have to rely on Christ's grace (cf. Aug. *c. ep. Pel*. 3, 16/504, 24–505, 24; *Ep*. 187, 29–31/106, 3–110, 4).

Christ is not only exceptional in regard to his human nature but also in regard to his divine being. On the one hand, Christ alone demonstrates and manifests the salvific operation which the whole trinity inseparably operates. On the other hand, God the Son is the only person of the trinity who became incarnate (cf. Aug. *trin*. 15, 20/489, 73–79). Due to this exceptionality, Christ is the *mediator* and the *reconciliator*, in contrast to the Father or the Holy Spirit. However, God the Father and God the Holy Spirit enduringly cooperate in Christ's work as mediator and reconciler.[24]

Augustine localised the act of reconciliation in Christ's redeeming death. He framed his Christological excursus with references to Christ's sacrifice (Aug. *ench*. 10, 33-13, 41), quoting first, Rom 5:9–10 (Aug. *ench*. 10, 33) and later, 2Cor 5:20–21 (Aug. *ench*. 13, 41)— both references play a substantial role in the Christocentric soteriology of the anti-Pelagian oeuvre (cf. Aug. *c. ep. Pel*. 3, 16/505, 12–15; 4, 8/528, 17–529, 28). Therefore, the whole Christological argumentation in the *Enchiridion* aims to substantiate how Christ's death could redeem humankind (cf. especially Aug. *trin*. 13, 15–24, where Augustine raised the question: *Quid est reconciliati per mortem filii eius* [Rom 5:10]? [*trin*. 13, 15/401, 3]). In the *Enchiridion,* Augustine did not attribute a specific role in the act of reconciliation to God the Father (*pater*) (cf. in contrast thereto, Aug. *trin*. 13, 15/402, 14–15: *In illa moritur pro nobis filius, et reconciliatur nobis pater per eius mortem*). In the *Enchiridion,* Augustine also omitted the image that God the Father sent Christ (cf. Aug. *Io. eu. tr*. 111, 1/628, 5–21; *c. Max*. 1, 2/494, 14–19).[25] Thus, Augustine did not primarily present the reconciliation of humankind with God (*reconciliari Deo*) as an act of Christ towards God the Father but as a process in Christ and for this very reason in the triune God himself. Because the divine and the human identity in Christ communicate and correspond with each other, during Christ's sacrifice an exchange between God and humankind occurs: Christ who is God humiliates himself and becomes a human. Christ, the human, who is full of grace and, thus, without sin, unjustly dies (cf. Aug. *ench*. 28, 108/107, 66–108, 83; cf. also *Ep*. 187, 20/99, 7–10). In this manner, God in Christ sympathises with fallen mankind. Therefore, those who are justly condemned to death by their sinful condition and habit receive justification through Christ's grace. This exchange between divine and human nature in Christ proceeds within all Christians who participate in Christ's destiny by grace (cf. Rom 6) (Aug. *ench*. 14, 52–53/76, 46–78, 103). However, not only Christ but the whole trinity operates and distributes grace. Therefore, Christians become the temple of God (*templum . . . Dei hoc est totius summae trinitatis* [Aug. *ench*. 15, 56/80, 40–41; cf. *Ep*. 187, 16–21/93, 18–100, 5, where Augustine differentiated between God's ubiquity in his creation (*esse*) and his gracious inspiration of Christians (*habitare*)]). Eschatologically, Christians will be resurrected as Christ has risen from the dead (Aug. *ench*. 14, 53/78, 103–111).

By describing Christ as *mediator* and *reconciliator*, Augustine introduced a Christology with an asymmetric dynamic between the two natures. In Christ, God and man communicate with each other and cooperate—although the divine part (Christ's divine nature and

God's grace) is always preceding and the human part always cooperating.[26] According to Augustine, the dynamic in Christ does not only correspond with the salvific dynamic between God and humankind. The dynamic in Christ is causal and pivotal for the salvific dynamic between God and humankind (cf. Madec 1989, pp. 287–312).

In contrast to his approach in the *Enchiridion,* Augustine did not emphasise the intra-Trinitarian operations during Christ' incarnation in his anti-Pelagian works with the exception of *De praedestinatione sanctorum.* He even mostly omitted explicit references to the two natures doctrine in his anti-Pelagian oeuvre. However, he pursued the same argumentative goal: to demonstrate that salvation is only achieved through the dynamic in Christ's incarnation and death. In *Contra duas epistulas Pelagianorum,* Augustine explicitly attacked the Pelagians for denying this salvific dynamic in Christ between God and humanity (Aug. *c. ep. Pel*. 1, 39/456, 14–457, 13; 3, 16/504, 17–505, 24). Thereby, he mainly focussed on Christ's human nature due to the controversy regarding the question of how Christ could become human without sin (cf. Rom 8:3) (cf. Lamberigts 2005; Keech 2012; Janssen 2023).

## 4. Defending a Christocentric Soteriology through a Trinitarian Christology

Several scholars have detected an imbalance in the relationship between Augustine's Trinitarian doctrine and his teaching of the economy of salvation (cf. for an overview Kany 2007, pp. 152–53, 194–95). Especially, Augustine's anti-Pelagian argumentation could be criticised due to this imbalance. On the one hand, Augustine emphasised the unity of God's operation: God is creator and saviour, he predestinates by grace and judges with righteousness.[27] On the other hand, Augustine declared the doctrine of the exceptionality of Christ to be the standard of orthodoxy.

For several reasons, his anti-Pelagian soteriology might be regarded as Christocentric. Augustine omitted explicit references to his Trinitarian doctrine throughout his anti-Pelagian oeuvre. Even in the *Enchiridion,* the Trinitarian structure given by the Creed fades into the background. Furthermore, Augustine propagated a Christocentric piety: Christians are saved by participating in Christ's fate (cf. Harmless 1995, pp. 375–82; Studer 2005, pp. 209–32; Ployd 2015, p. 71). Thus, Christ is the mediator and the redeemer in person, not an abstract deity. Therefore, the Numidian bishop emphasised Christ's incarnation and, especially, his death as the only means of salvation. In an anti-Modalist manner, he argued that only the Son incarnated and died (cf. Aug. *praed. sanct*. 2, 67/270, 1–271, 30).

Augustine's statement *inseparabilia sunt opera trinitatis ad extra* implicates the unity of God's operation, however, as a cooperation of God the Father, God the Son and God the Holy Spirit. The Christology demonstrates that according to Augustine, the three persons of trinity always act individually and always cooperate inseparably (cf. Aug. *trin*. 13, 15/402, 23–24; 15, 20/486, 1–489, 84; *praed. sanct*. 1, 13–15/191, 1–194, 29; *c. s. Arrian*. 9/206, 51–224, 39) (cf. Bailleux 1971, pp. 189, 217–18). Thus, according to Augustine, the Trinitarian and the Christocentric approach do not disagree. They are rather two perspectives on the same subject: a Trinitarian soteriology through Christ as well as a Christocentric soteriology through the trinity.

However, in the majority of Augustine's anti-Pelagian works, the Trinitarian perspective on soteriology is absent (cf. Section 3.1), although he argued for the cohesion of God's operation as creator and saviour (cf. Section 3.3). Instead, Augustine emphasised mostly the Christological aspects of soteriology and on occasion the Pneumatological (cf. Section 3.2). In regard to these aspects, the *Enchiridion* resembles the emphasis of nearly all anti-Pelagian works. However, in the *Enchiridion* and in some other writings after 420, such as *De praedestinatione sanctorum,* Augustine also presented a Trinitarian Christology (cf. Section 3.4). This observation leads to a subsequent question: why did Augustine begin to embed his Christological soteriology in his Trinitarian doctrine (only) during the later Pelagian controversy?

In his early anti-Pelagian writings, beginning with *De peccatorum meritis,* Augustine emphasised Christocentric soteriology. Throughout the controversy, Augustine never changed this prioritisation radically. The main reason for this emphasis is that the Pelagian

controversy was in a large part a controversy on Christ-centred piety. The controversy revolved around several questions about Christ's work and person: How has Christ died for the sake of a newborn? How and to which degree can humans imitate Christ? What is participation in Christ, especially, concerning baptism, prayer and Eucharist? Against the so-called Pelagians, Augustine aimed to demonstrate that Christ's grace radically transforms humans so that they are reconciled with God. Augustine attacked the Pelagians for denying this insight. Therefore, he called them *inimici gratiae Christi* (Aug. *c. ep. Pel.* 1, 9/430, 25), respectively, *inimici gratiae Dei quae uenit per Iesum Christum* (Aug. *c. ep. Pel.* 1, 42/458, 26–27).

In the *Enchiridion*, Augustine transferred this insight into a catechetical context. For example, he emphasised the necessity of baptism for adults and newborns (Aug. *ench.* 13, 42-14, 55) and stated that the life of a Christian is an enduring participation in Christ's incarnation and death (Aug. *ench.* 14, 53/78, 97–111). Moreover, Augustine demonstrated that even Christ's human nature is reliant on grace. Thus, Christ's incarnation symbolises that every human has need of God's grace. With this emphasis on participation in Christ, Augustine responded to Laurentius's request to outline the fundamentals of the Christian faith (Aug. *ench.* 1, 4/50, 34–36) by stating that first and foremost faith in Jesus Christ is necessary for salvation. Thus, in the *Enchiridion*, Augustine presented a Christocentric piety which manifests itself in faith, baptism and prayer.

However, the *Enchiridion* also illustrates that Augustine (progressively) embedded his Christocentric soteriology into his Trinitarian doctrine during the Pelagian controversy. Thereby, he strengthened and defended his Christocentric soteriology with a Trinitarian explanation.

In *De spiritu et littera,* Augustine combined his Christological approach from *De peccatorum meritis* with a Pneumatological one. This Pneumatological soteriology is complementary to the Christological approach. By focussing on the Christological soteriology, Augustine especially emphasised the relationship between God and humans. While arguing that Christ's sacrifice is the only means of salvation, Augustine often applied to the reconciliation between God and humans. With his Pneumatological approach, Augustine tried to present more clearly how God's grace could transform the human mind (cf. Yam 2019, pp. 448–50, 606–13). The reference to the Holy Spirit enabled Augustine to gain a deeper understanding of the hidden operation of grace within the human mind. It is hardly surprising that Augustine referred to this Pneumatological soteriology again in *Contra duas epistulas Pelagianorum* (Aug. *c. ep. Pel.* 3, 11–12/497, 10–499, 27) when his understanding of grace was attacked as a deterministic concept. Augustine responded that grace is not forcing humans to believe against their will, but the Holy Spirit persuades humans by grace.

While combining a Christological and a Pneumatological approach to soteriology, Augustine obviously premised the approval of a Trinitarian doctrine. However, he did not explicitly refer to his Trinitarian doctrine (cf. Aug. *spir. et litt.* 59/219, 13–25). Instead, he mainly presented God the Son and God the Holy Spirit as two operating persons whose operations refer to each other (cf. Section 3.2: Trinitarian "appropriations"). His main argument, thereby, was that the Holy Spirit arouses the faith in Christ's salvific operation (cf. Aug. *spir. et litt.* 15/167, 13–23; 25/179, 5–7; *c. ep. Pel.* 3, 11/497, 19–20). In this context, Augustine did not emphasise the intra-Trinitarian bonds between God the Son and God the Holy Spirit.

In contrast to this approach to combine Christology and Pneumatology merely on the basis of soteriology and not on the basis of the doctrine of God, Augustine explicitly amalgamated his anti-Pelagian Christocentric soteriology with a Trinitarian doctrine in the *Enchiridion*, in *De trinitate* 13–15 and later, in *De praedestinatione sanctorum*. Therefore, he pursued the question of how the triune God operates in Christ's incarnation and death (cf. Section 3.4: Trinitarian Christology). Beforehand, Augustine had only focussed on the operations of specific persons of the trinity while presenting his anti-Pelagian

soteriology. Augustine felt obliged to enhance his anti-Pelagian argumentation explicitly by his Trinitarian doctrine, around 420, due to several reasons.

During the midst of the Pelagian controversy, Augustine increasingly introduced his anti-Pelagian argumentation into other discussions and contexts, including his Trinitarian doctrine of God. The *Enchiridion*, *Ep.* 187 or *De trinitate* 13 are only a few examples of this endeavour. In contrast thereto, even during the later Pelagian controversy, Augustine seldom referred to his Trinitarian doctrine in works which were polemical and directed against his "Pelagian" opponents such as Julian. Moreover, around 420, Augustine learned that "Arian" Homoians entered Africa and propagated their faith. For apologetic reasons, Augustine had to emphasise that Christ's incarnation and death did not imply a subordination of Christ (cf. Aug. *praed. sanct*. 2, 67/270, 1–271, 30 where Augustine explicitly refuted Arianism, Apollinarianism, Manichaeanism and Photinianism). Ensuing from these arguments, one could easily conclude that Augustine began to reflect his anti-Pelagian soteriology by his Trinitarian doctrine mainly due to discussions on themes which had nothing in common with the Pelagian controversy and Augustine's anti-Pelagian argumentation.

However, Augustine's increasing engagement with how he could combine his Trinitarian doctrine with his anti-Pelagian soteriology should also be read in the light of the development of the Pelagian controversy.

Already, before 420, Augustine had introduced his anti-Pelagian soteriology into other contexts and discussions in which his Trinitarian doctrine played a major role (cf., for example, *De trinitate* 8–10 (cf. Yam 2019) or *De ciuitate Dei*), however, without emphasising a Trinitarian Christology. Moreover, Augustine chose, in the *Enchiridion*, to interpret the Creed in a new fashion. He reduced the presentation of the intra-Trinitarian doctrine, however, and highlighted a Trinitarian reflection on Christology and soteriology. Furthermore, he applied to this Trinitarian Christology while debating with the monks of Hadrumetum and Southern Gaul. Thus, Augustine's Trinitarian Christology, which he presented in the *Enchiridion*, primarily engages theological problems of the Pelagian controversy (and does not primarily refer to themes from discussions with Homoians or other antitrinitarian heresies).

In Section 3.4, we have already observed that Augustine referred to the intra-Trinitarian operations in Christ's incarnation to emphasise Christ as an example of grace (cf. McWilliam Dewart 1982; Kantzer Komline 2020, pp. 299–308). Augustine employed this argument to refute opponents who claimed that the beginnings of faith (*initum fidei*) could be understood as a cooperation between God and humans in which humans could already begin to believe independently from God's grace. In the *Enchiridion*, Augustine stated explicitly that Christ's human nature did not earn the union with God the Son. This union was granted by grace (Aug. *ench*. 11, 36/70, 13–30; cf. *praed. sanct*. 1, 31/206, 2–5). Grace also operated that Christ's human nature would never fall into sin as Adam has done (cf. Aug. *corrept*. 30/254, 1–255, 38).

However, the Trinitarian approach to Christology in the *Enchiridion* was not only about anthropological–soteriological questions as McWilliam Dewart (1982) claims. Augustine reemphasised the exceptionality of Christ's human nature (cf. Section 3.4) which the Pelagians would deny according to his opinion (cf. Aug. *c. ep. Pel*. 2, 3/462, 28: *Pelagiani autem carnem redimendorum carni redemptoris aequando*). Furthermore, by interpreting Christ's incarnation as an inseparable operation of the trinity, Augustine reacted to two other theological problems of the Pelagian controversy which are related to the doctrine of God.

A main topic of the Pelagian controversy was the relationship between God's creational and saving operations. Augustine attacked the Pelagians for neglecting God's redeeming acts by equalising them with God's creational acts (cf. Janssen forthcoming, chp. 3.7). Therefore, Augustine focussed on Christ as saviour. However, his own theology was attacked for depreciating God as creator (cf. Julian, *Ep. ad Rufum fr.* 1/336, 3–337, 12; 12/338, 54–64). Augustine answered that the salvation by Christ surpasses the creation of the triune God, however, not as contradiction but as redemption and consummation (cf. Janssen forthcoming, chp. 9.3.2). In this context, Augustine referred to the Pneumatology

(cf. Aug. *spir. et litt*. 5–6/157, 10–159, 4; 13/165, 18–166, 3; 25–27/179, 1–181, 22; 66/228, 3–229, 12) and to his Theocentric doctrine of creation (cf. Aug. *c. ep. Pel*. 4, 4/524, 1–525, 11; *nupt. et conc*. 2, 48–50/303, 3–307, 14). In the *Enchiridion*, Augustine enhanced this strategy. On the one hand, he argued in the *Enchiridion*, as he had argued several times before in his anti-Pelagian works, that the triune God is likewise the creator and saviour without distinguishing the persons of the trinity (cf. Section 3.3). On the other hand, he unfolded in detail how the salvation by Christ is an operation of the whole trinity. Thereby, he connected salvation and creation in the person of Christ whose human nature is a new creation by grace (Aug. *ench*. 11, 38/71, 30–35). According to the *Enchiridion*, Christ's incarnation demonstrates that the triune God is the creator and saviour. With this reference to Christ's incarnation, Augustine tried to reject the charge of Manichaeism. Moreover, he attacked the Pelagians for denying the exceptionality of Christ's person and salvific work as mediator and reconciler.

Furthermore, around 418, Augustine's theory of predestination became publicly criticised. In his earlier anti-Pelagian works, Augustine had only treated his theory of predestination as the logical conclusion of his soteriology (cf. Aug. *pecc. mer*. 1, 29–30/27, 22–29, 16). Especially in the 420s, the doctrine of predestination became a distinct topic in the debate because Augustine's opponents, especially Julian of Aeclanum and later monks from Hadrumetum and southern Gaul, directly attacked Augustine for advocating the image of a tyrannical and cruel God who does not will to save all humans. According to his critics, the merciful God in Christ has nothing in common with Augustine's almighty God who predestinates and condemns (cf. Ogliari 2003, pp. 305–402). Augustine himself raised this question in *Enchiridion* 27, 103 (105, 26–29):

> *Quid est enim eorum unde non Deus per unigenitum suum dominum nostrum per omnes gentes saluos homines fieri uelit et ideo* faciat, *quia omnipotens uelle inaniter non potest quodcumque uoluerit?*

While responding, Augustine aimed to demonstrate that God's predestination, mercy and judgement are complementary and not contradictory. In the *Enchiridion*, Augustine mainly argued by interpreting 1Tim 2:4. He claimed that God mercifully distributes grace in Christ (Aug. *ench*. 24, 97-28, 108) and judges with righteousness in Christ as well (Aug. *ench*. 29, 110). Augustine reinforced this argument in *De praedestinatione sanctorum* by connecting his doctrine of predestination with his Trinitarian Christology: Christ is exceptional gifted by grace (cf. already Aug. *ench*. 10, 35/69, 55; 11, 36/70, 27) and predestinated (Aug. *praed. sanct*. 1, 30–31/205, 1–207, 36). Christ is also electing those to whom he distributes grace. Thus, those who receive grace receive grace in Christ; those who are predestinated are predestinated in Christ (cf. Eph 1:4 [Aug. *praed. sanct*. 1, 34–35/210, 1–211, 16]).

Although Augustine did not conceptualise the *Enchiridion* as a polemical and explicit anti-Pelagian work, he enhanced his anti-Pelagian argumentation in the *Enchiridion*. Since 418/19, his anti-Manichaean fire wall that God is good creator and saviour was shattered by the criticism of Julian of Aeclanum. Moreover, Augustine's theories of grace and of predestination became major subjects in the theological discourse. In response, Augustine did not desert his Christological approach to soteriology; instead, he referred to his Trinitarian doctrine which he contemporaneously developed further in *De trinitate* 13–15 to defend his anti-Pelagian Christocentric soteriology. By stating that the incarnated Christ is God the Son, as well as the product of the creational and gracious operation of the triune God, Augustine connected the different operations of God—as distributer of grace and as judge, as creator and as saviour—in the person of Christ. With this Trinitarian Christology, Augustine also employed Christ's incarnation as a prime example of God's merciful operation and of the character of grace. By referring to his Trinitarian doctrine, Augustine tried to substantiate his anti-Pelagian core thesis: salvation is only achieved through Christ's sinless sacrifice.

Thus, Augustine did not coincidentally combine Trinitarian doctrine and anti-Pelagian soteriology in the *Enchiridion* because he had to respond to Laurentius's request for a Christian handbook. His approach in the *Enchiridion* is a thoughtful anti-Pelagian interpretation

of the Creed which introduces aspects of the Trinitarian doctrine into the anti-Pelagian soteriology for apologetic reasons. With this approach to combine Christology, Trinitarian doctrine and soteriology, Augustine offered a solution path to some of the most urgent critical inquiries to his (anti-Pelagian) doctrine.

**Funding:** I acknowledge support by Open Access Publishing Fund of University of Tübingen.

**Acknowledgments:** I give thanks to Han-Luen Kantzer-Komline for very helpful remarks.

**Conflicts of Interest:** The author declares no conflict of interest.

## Notes

1   For Augustine's anti-Pelagian argumentation, I will focus on *Contra duas epistulas Pelagianorum* because Augustine wrote this work at about the same time as the *Enchiridion* (420/21). Furthermore, Augustine outlined in *Contra duas epistulas Pelagianorum* nearly his whole anti-Pelagian argumentation (cf. Janssen 2023; Janssen forthcoming).

2   The most influential interpretation of the *Enchiridion* was produced by the German theologian Adolf von Harnack (Von Harnack 1910, pp. 220–36), who employed the *Confessiones* and the *Enchiridion* to outline Augustine's theology. Harnack chose the *Enchiridion* for this endeavour due to two reasons (ibid., pp. 101–2): (1) He claimed that Augustine has fostered the ecclesiological dogma instead of the Christological. As a core element of this ecclesiological dogma, Harnack emphasised Augustine's interpretation of the Creed. (2) Harnack focussed on Augustine's anti-Pelagian soteriology as the innovative culmination of his theology (ibid., p. 232). According to Harnack, the *Enchiridion* demonstrates both the innovational strength and the inconsistency of Augustine's theology ("alles vereinigt sich an diesem Buche, um uns darüber zu belehren, worin die Umstimmung (und andererseits die Verstärkung) der vulgär-katholischen dogmatischen Lehre durch Augustin bestand." [ibid., p. 220]). Harnack concluded that although Augustine committed himself to the traditional Creed, the African Church father partially overcame the intellectualism and dogmatism of the catholic Church by focussing on grace and (individual) piety (ibid., pp. 231–36). Following Harnack, Paul Simon called the *Enchiridion* in the introduction to his German translation of Augustine's first systematic treatise ("erste[] Versuch einer systematischen Darstellung der wichtigsten Glaubenslehren") (Simon 1962, p. 8).

3   That, with his *Enchiridion,* Augustine did not produce his final dogmatic draft, even Harnack himself had to admit (Von Harnack 1910, p. 99). Instead, Augustine aimed to outline the most important aspects of the Christian faith which is required to obtain eternal salvation (Aug. *ench*. 1, 5/50, 48–67). Therefore, his approach is highly selective. In the *Enchiridion,* he focussed on specific soteriological questions while neglecting other topics which are central for his theology, for example, the role of the Old Testament and the law (cf. the short notes in Aug. *ench*. 31, 118/112, 25–44), the origin of the soul and most parts of his metaphysical–ontological system. Furthermore, Augustine did not explicitly refute heresies such as Pelagianism or Manichaeism in the *Enchiridion* (Aug. *ench*. 1, 5–6/50, 57–51, 75). Thus, Harnack's appreciation of Augustine's theology in the *Enchiridion* because the African Church father did not dedicate himself to speculative theology (i.e., cosmology, extensive considerations of Trinitarian and Christological doctrine) in this work (Von Harnack 1910, pp. 99, 233), merely reveals Harnack's own theological stance. Cf. also Scheel (1937, chp. v): "Tatsächlich bietet das Enchiridion nur einen im Zusammenhang anderweitiger Ausführungen Augustins vollständig zu verstehenden Ausschnitt aus seiner Gedankenwelt und in diesem Ausschnitt eine eigentümliche Vermengung halb evangelischer Motive mit katholischen, z.T. ,vulgär-katholischen' Elementen."

4   Cf. this argument in Aug. *trin*. 12, 22; 14, 1 (375, 19–29; 421, 1–15); *spir. et litt*. 18 (170, 7–171, 11); *ciu*. 10, 1 (273, 67–274, 100); *Ep*. 167, 11 (598, 3–599, 11).

5   Cf. Aug. *trin*. 12, 22 (375, 25–27): *Et quis cultus eius [= Dei] nisi amor eius quo nunc desideramus eum uidere credimusque et speramus nos esse uisuros.*

6   In this context, Augustine referred to *c. mend*. (Aug. *ench*. 6, 18/58, 1–4). In this work, Augustine observed that a moral relativism which exculpates lies can lead to an epistemological relativism (cf. Fürst 2004–2010a, p. 1264; Fürst 2004–2010b, pp. 1267–70). As a consequence, every search for truth would be doomed to fail and the fundament of faith would implode.

7   Cf., for example, the argumentation in *ench*. 10, 30–32: Augustine combined 2Petr 2:19, Rom 6:20–22 (as a paraphrase) and Joh 8:36 claiming that the postlapsarian will is either addicted to sin or to justice (*ench*. 10, 30/65, 33–66, 57). Therefore, the will is only free when it is freed from sin. This argumentation can be found in *c. ep. Pel*. 1, 5 (425, 21–426, 28). Furthermore, Augustine argued with Rom 9:16, Phil 2:13 and Ps 50:12 that God enables humans to believe (*ench*. 10, 31–32/66, 65–67, 97). Augustine had already combined Rom 9:16 and Phil 2:13 in *Simpl*. 1, 2, 12 (37, 334–344) to demonstrate this thought (cf. *c. ep. Pel*. 1, 36/453, 4–8, however, with Prov 8:35 instead of Ps 50:12—although Prov 8:35 is missing in *ench*. 10, 32. Augustine referred to this verse at this passage [Aug. *ench*. 10, 32/67, 98–99]: *Qui [= Deus] hominis uoluntatem bonam et praeparat adiuuandam et adiuuat praeparandam*). Ps 50:12 appears, for example, in *c. ep. Pel*. 3, 6 (492, 10–11) to describe God's gracious operation. Finally, Augustine quoted Ps 22:6 and 58:11, stating that God's grace precedes and follows the will (Aug. *ench*. 10, 32/67, 101–110): The will can only be good due to grace, moreover, the will can only remain good by grace. Augustine presented the same argumentation in *c. ep. Pel*. 2, 21 (482, 28–483, 18). Even the example of praying for one's own enemies can be found in *c. ep. Pel*. 1, 37 (454, 5–10).

[8] Augustine employed two types of his anthropological–soteriological stages which are complementary to each other: The first scheme highlights the difference between the prelapsarian (*ante peccatum*) and postlapsarian stage (*post peccatum*). The second scheme emphasises the role of the law by differentiating the stage *sub peccato* into *ante legem* and *sub lege*. Although Augustine quoted the second scheme in *ench*. 31, 118, he structured the *Enchiridion* according to the first scheme as he nearly omitted the salvation history and the role of the law throughout the work.

[9] Cf. Schindler (1996–2002, pp. 1313–14). Augustine emphasised the Trinitarian doctrine in *De fide et symbolo* far more than in *ench*. His main motive in *De fide et symbolo* is to refute heresies which are teaching against the Creed (Aug. *f. et symb*. 1/4, 11–15).

[10] In this sermon on the Creed, Augustine proceeded verse by verse. Thus, Augustine began with God the Almighty creator (Aug. *symb. cat*. 2/185, 16–186, 51). Afterwards, he discussed the consubstantiality of God the Father and God the Son (Aug. *symb. cat*. 3–5/186, 52–189, 131). Then, he outlined Christ's salvific incarnation and death. Thereby, Augustine focussed on the merits of Christ's incarnation and death for the believers (Aug. *symb. cat*. 6/189, 137–138: *Humilitas Christi quid est? Manum Deus homini iacenti porrexit*; *symb. cat*. 9/192, 212–214: *Ostendit nobis, in cruce quid tolerare, ostendit in resurrectione quid sperare debeamus*.). Moreover, he encouraged them to endure in faith. A main difference between this sermon (*symb. cat*.) and the *Enchiridion* is its "*Sitz im Leben*". With this sermon, Augustine aimed to explain the meaning of the Creed to catechumens (Harmless 1995, pp. 274–86). He especially exhorted them to memorise the Creed and to believe in their hearts (Rom 10:10) (Aug. *symb. cat*. 1/185, 1–15).

[11] Further similarities between the *Enchiridion* and *De trinitate* can be detected. Both works combine a Trinitarian and Christological doctrine with soteriology (cf. Madec 1986–1994, pp. 883–90; Studer 2005, pp. 171–79). In both works, Augustine aimed to guide his readers in the right ways to θεοσέβεια, i.e., the knowledge and adoration of God through faith and love. Therefore, Augustine distinguished between faith and the vision of God (cf. Aug. *ench*. 1, 5/50, 48–53; 2, 8/52, 37–53; 16, 63/83, 47–62; *trin*. 13, 26/418, 27–420, 77; 15, 14/479, 1–480, 41). Moreover, the epistemological excursus in *ench*. 5, 16–8, 23 resembles thoughts from *De trinitate* 15 (cf. Brachtendorf 2000, pp. 266–81).

[12] While speaking of *Deus pater omnipotens*, Augustine referred to formulations of the Creed; sometimes, he even paraphrased the Creed (cf. Aug. *ench*. 24, 96/100, 38–40: *Nostrae confessionis initium, qua nos in Deum patrem omnipotentem credere confitemur*).

[13] In contrast thereto, Basil Studer argues that Augustine distinguished between *Deus proprie* which explicitly denotes God the Father and *Deus communiter* which signifies the triune God (Studer 1993, pp. 188, 194, 262–63, 265). For comparison, see *f. et symb*. 2–3 (4, 16–6, 22) where Augustine referred to the first part of the Creed (*in Deum patrem omnipotentem*). Afterwards, he named God, the Almighty (*Deus omnipotens*), the creator. Thus, Augustine did not explicitly attribute the act of creation to God the Father. By naming Christ as the agent of creation (*quia omnia per uerbum creauit*), Augustine, however, conveyed the impression that God the Father is the root of creation.

[14] Augustine only employed the term *uerbum* for God the Son recurring to Joh 1:14 (*uerbum caro factum est et habitauit in nobis*) (Aug. *ench*. 10, 34–11, 36).

[15] Augustine focussed on the soteriological role of Christ from *ench*. 10, 33 onwards. However, he already inserted Christological interpretations of the soteriology beforehand: the liberation of the will is achieved by Christ (Joh 8:36) (Aug. *ench*. 10, 30/66, 50–53), humans become in Christ a new creature (Eph 2:10; 2Cor 5:17) (Aug. *ench*. 10, 31/66, 62–69).

[16] In his doctrine of grace, Pelagius also focussed on Christ's salvific death (cf. Dupont 2006, pp. 359–60). However, the effect of Christ's death was disputed between Augustine and Pelagius. Augustine harshly criticised the *Pelagiani* for denying that Christ's death was an atonement for the original sin (*peccatum originale*) (Aug. *c. ep. Pel*. 1, 12/432, 9–433, 2; 1, 39/456, 14–457, 13).

[17] Ployd (2015), demonstrates how Augustine employed his Trinitarian doctrine in his anti-Donatist argumentation: Augustine especially focussed on the unity of the triune God as fundament of the unity of the Church. In the anti-Pelagian argumentation, Augustine also referred to the unity of God's operation, however, not concerning ecclesiology, but soteriology. Thus, Augustine's emphasis in his anti-Donatist and his anti-Pelagian works slightly differs; however, Augustine focussed in both controversies on the unity of God's operations *ad extra*.

[18] Cf. for the anti-Arian argumentation in *De trinitate* Barnes (1993); and Ayres (2010, pp. 171–73).

[19] It is conspicuous how often Augustine referred in the *Enchiridion* to the angels. He stated that certain angels were fallen due to their own will (Aug. *ench*. 4, 14–15; 8, 26–28), that God refills the number of the fallen angels with those humans who are saved (Aug. *ench*. 9, 29) and that the Church currently consists of angels and redeemed humans (Aug. *ench*. 15, 58–16, 62). Thereby, Augustine even argued that Christ did not die for the good angels because they never deserted God (Aug. *ench*. 16, 61/82, 24–25). Augustine mostly focussed on angels in his anti-Manichaean and anti-Platonic works; in his anti-Pelagian oeuvre, the angelology is neglected. The emphasis on the angels demonstrates that the *Enchiridion* is not directed against the Pelagians alone. Instead, Augustine tried to outline in the *Enchiridion* especially his soteriology which had, after 411, an anti-Pelagian character.

[20] Augustine often combined the charge of introducing two sons (*duo filii*) instead of the one Christ with the accusation of rendering the *trinitas* in a *quaternitas* (cf. Aug. *praed. sanct*. 1, 31/206, 16–207, 19; 2, 67/270, 7–20; *c. s. Arrian*. 6/194, 25–195, 47). On the one hand, this charge distinguished his views from Apollinarianism and Arianism (cf. Drobner 1986, pp. 261–62). On the other hand, Augustine reacted to theological ideas of some (for example, the Gaul Leporius) who strictly distinguished between the two natures of Christ (cf. Krannich 2005, pp. 54–71).

21    In the *Enchiridion,* Augustine did not emphasise the emotions ("vie affective") of Christ's human nature (cf., therefore, van Bavel 1954, pp. 119–75).

22    Augustine emphasised the unity of Christ's person by comparing the two natures in one person with the composition of a human from soul and body (*ench.* 11, 36/69, 8–70, 13). Furthermore, Augustine referred back to *Ep.* 137 (*ench.* 10, 34/69, 45–47) in which he had even stated that Christ is a mixture (*persona . . . Christi mixtura*) of God and human such as the human is a mixture of soul and body (Aug. *Ep.* 137, 11/109, 15–111, 7) (cf. Drobner 1986, pp. 169–171, 252–53). Especially in his anti-Pelagian works, Augustine stressed that both soul and body have their own force and operation. According to Augustine, the soul should reign over the body, but in the sinner, the passions (*concupiscentia*) of the body dominate the soul. Thus, Augustine's image of the soul–body mixture for Christ's two natures does not contradict his dyotheletic Christology. Compared to *Ep.* 137, Augustine was, however, more emphatic about the two identities in Christ in the *Enchiridion.*

23    Although Augustine's "Christology of grace" (McWilliam Dewart 1982) shows a resemblance to the "Antiochene" Christology (cf. Von Harnack 1910, pp. 126–34, one cannot describe Augustine as an "Antiochene" theologian. These approaches overestimate the differentiation between a hypostatic union and a union by grace in Augustine's Christology (cf. McGuckin 1990, pp. 47–50). According to Augustine, the Christological union is personal (*unitas personae*): the pre-existent Son who is grace by nature assumes a human by grace. Moreover, Theodor of Mopsuestia harshly criticised an Augustinian anthropology which is inextricably bound with his Christology (cf. Malavasi 2015).

24    Dodaro 2010, gives an insight in the scholarly debate regarding whether Augustine regarded Christ or the Holy Spirit as the central person in the mediation of grace. According to Verhees (1976, p. 253), Augustine characterised the Holy Spirit as the moving force in the act of Christ's incarnation. However, Verhees underestimates the personal union between God the Son and a human.

25    Since 418, Augustine learned that his opponents tried to solve the antithesis of grace and merits by stating that the impulse of faith is a voluntary act of humans. According to them, God's grace supports those who want to believe. According to Augustine, his opponents, thereby, defined faith to be a human merit (cf. Aug. *Ep.* 194, 9/183, 4–5). Augustine responded that nobody could believe in Christ who has not received grace from God. Thereby, he referred to Joh 6:66[Vulg] ( . . . *nemo potest uenire ad me, nisi fuerit ei datum a patre meo*) to prove that everybody requires God's grace. Augustine employed a similar argument in *Contra duas epistulas Pelagianorum* 1, 6 (427, 20–428, 16). In contrast thereto, he also dismissed in *Epistula* 194 a possible misunderstanding of this verse by emphasising the inseparability of God's operations *ad extra*. Moreover, he declared that Christ's incarnation is a prime example of humility (Aug. *Ep.* 194, 12/185, 16–186, 13). Augustine revisited this topic in *De praedestinatione sanctorum* 1 (cf. Aug. *praed. sanct*. 1, 9/187, 1–188, 32). Augustine quoted Joh 6:29 and 6:43–45[Vulg] to support his doctrine that God's grace is evoking faith. However, Augustine inserted an excursus to prevent an "Arian" misunderstanding of the Son's sending by God the Father: Joh 6 does not imply a subordination of the Son. Instead, the triune God inseparably arouses faith in the believers (Aug. *praed. sanct.* 1, 13/191, 1–192, 29).

26    Cf. for Augustine's concept of a *communicatio idiomatum* (Drobner 1986, p. 258; van Bavel 1954, pp. 57–63). Due to this dominance of the divine nature, several studies accuse Augustine of Miaphysitism (Geerlings 1978, pp. 256–58) or even Apollinarianism (cf. Scheel 1901, pp. 77, 210–74; Geerlings 1978, pp. 105–12, 122–24, 137, 145; Keech 2012, pp. 181–89). Just as the identification of Augustine's Christology with the "Antiochene" approach (cf. note 23), this interpretation is one-sided: it neglects Augustine's emphasis that Christ had a human will. Moreover, Augustine did not think in the categories of the Christological controversy; thus, his Christology should not be judged with these categories. Instead, one should regard Augustine's Christology in the context of late-antique Latin Christology which had, for example, only a rudimentary understanding of Apollinarianism. The main Christological debates in which Augustine participated were with Platonists and Manichaeans (how could God become incarnate?) and Homoians (how can the incarnate remain God?). Furthermore, Augustine discussed especially Christ's human nature with ascetics (Jovinian) and during the Pelagian controversy.

27    In contrast thereto, Basil Studer claims that Augustine's soteriology has to be Theocentric by structure because Augustine emphasised the unity of God's creational and redeeming operation (Studer 1993, pp. 282–83; cf. Geerlings 1978, pp. 62–63, 69–77). Studer argues that according to Augustine, God redeems humankind through Christ (*per Christum*) (Studer 1993, pp. 117–18, 238). Moreover, Studer detects a slightly "unitarian tendency" in Augustine's Trinitarian doctrine referring to the concept that *inseparabilia sunt opera trinitatis ad extra* (Studer 2005, pp. 181–89, 194; cf. Dodaro 2010), although Studer also admits that Augustine was convinced of the singularity of the three Trinitarian persons (Studer 2005, pp. 204–08).

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
