# Peer review of "Augustine’s Enchiridion: An Anti-Pelagian Interpretation of the Creed"

_religions, doi:10.3390/rel14030408_

Round 1

Reviewer 1 Report

This article offers a thoughtful reading of the theological themes that intersect in a non-polemical, formational text. The author clearly demonstrates that Augustine incorporates Christological, trinitarian, and anti-Pelagian concepts into the Enchiridion. The article demonstrates thorough engagement with both primary and secondary literature. Although, I wonder how engagement with Ployd, Augustine, the Trinity, and the Church might help elucidate a similar type of move--i.e., showing how Augustine uses trinitarian principles in what is presumably not a trinitarian polemical context. That work does not consider Pelagianism or the Enchiridion, but it might provide a helpful precedent for the sort of work the present author is doing.

Regarding matters of argumentation, I think what the author puts forward could be just a tad clearer. In particular, the 'five theses' at the end are quite helpful in summarizing the argument and could be moved forward to help guide the argument and the reader's engagement with it. 

I think the best contribution is the author's refusal to given into the trinity-centric/christocentric binary and to show how these are necessarily mutually entwined themes for Augustine. I think this could be punched up even more, however, emphasizing this contribution. 

Author Response

Dear reviewer,

thank you very much for your helpful review.

This article offers a thoughtful reading of the theological themes that intersect in a non-polemical, formational text. The author clearly demonstrates that Augustine incorporates Christological, trinitarian, and anti-Pelagian concepts into the Enchiridion. The article demonstrates thorough engagement with both primary and secondary literature. Although, I wonder how engagement with Ployd, Augustine, the Trinity, and the Church might help elucidate a similar type of move--i.e., showing how Augustine uses trinitarian principles in what is presumably not a trinitarian polemical context. That work does not consider Pelagianism or the Enchiridion, but it might provide a helpful precedent for the sort of work the present author is doing.

I read Ployd's study and added two remarks in my paper. This study is very helpful to understand why Augustine focused on which aspect of his Trinitarian doctrine when he introduced it into his anti-Pelagian argumentation.

Regarding matters of argumentation, I think what the author puts forward could be just a tad clearer. In particular, the 'five theses' at the end are quite helpful in summarizing the argument and could be moved forward to help guide the argument and the reader's engagement with it. 

I have slightly restructured the article by moving four of the five theses forward to the beginning of chapter 3. Furthermore, I have changed the theses so that each thesis explicitly refers to one of the four sub-chapters (now: 3.1-4; in the first versions chapters 3-6).

I think the best contribution is the author's refusal to given into the trinity-centric/christocentric binary and to show how these are necessarily mutually entwined themes for Augustine. I think this could be punched up even more, however, emphasizing this contribution. 

I have enhanced the final sections of the article to demonstrate how Augustine connected his anti-Pelagian soteriology which often emphasises the role of Christ with a Trinitarian perspective. Thereby, I tried to strengthen my thesis that Augustine referred to his Trinitarian doctrine to substantiate and defend his anti-Pelagian soteriology.

Reviewer 2 Report

This article examines the interplay between trinitarian and anti-Pelagian themes in Augustine's Enchiridion.  It is in some ways difficult to assess.  On one side, the author knows the relevant primary and secondary material very well--it seems clear that the article is, to a considerable degree, drawn from work carried out for a book-length project--and is able to make intelligent identifications of some of the theological dynamics that play out in the text.  On the other side, it is not clear that the author has put his or her grasp of the material to work to develop a real original contribution to scholarship.  The commentary is offered almost entirely in a descriptive mode; a fair bit of the commentary--e.g., the noting of overlaps between Ench and other texts and cross-referencing to other scholarship--would be more at home as notes in a critical edition of Ench than in a journal article.  There is something of an 'of course-ness' to the claims that the author demonstrates--i.e., above all, *of course* both anti-Pelagian and trinitarian conceptions would appear and intersect with each other in Enchiridion.  Could we possibly expect Augustine to provide a handbook of Christian faith without drawing from the conceptions that he had spent years developing?

In the end, I think the article does enough to merit publication, particular in a special issue of a journal rather than as a standalone original piece.  A few thoughts that might be pursued in a revision:

1. Structurally, the article may try to do too much even as it risks doing too little.  The risk of doing too little resides in the descriptive 'of course-ness' of the enterprise.  The danger of doing too much resides in the range of topics that the essay treats.  Eight different sections in an essay of ten pages (in the electronic version available to me) is a lot.  Several of them are quite flat descriptions of what is in Ench with fairly workmanlike notes of overlaps and cross-references.  Might fewer sections with more original or critical reflection--sections 7 and 8 are the most obvious candidates for expansion--be richer?

2. The article takes on different tasks in sections 7 and 8 from the rest of the article.  A couple of thoughts: (i) how far does the question in section 7 fit the logic of Augustine's work?  The idea that a theology can be either trinitarian or Christocentric is distinctively modern; Augustine would not have thought that you could have one without the other, or that the two could possibly compete with each other.  Do we learn anything important by pressing these categories onto Augustine, or would it be more interesting to observe the ways in which he argues exegetically rather than deductively, i.e., he seeks in the first instance to interpret Scripture, and makes deductions from other doctrines primarily when they are needed to resolve ambiguities in the text, rather than supposing, as a matter of principle, that some one doctrine should have primacy, with others conceptions deduced from it.  (ii) It may be that the best version of this article would be structured around the five theses offered in section 8.  At present, these theses are presented in rather compressed, summary form, and differ in genre from the rest of the essay.  The essay as a whole would, however, be more obviously an original and analytic enterprise if it were structured by these five theses, with the descriptive material that otherwise appears in sections 1-7 reorganised as evidence in support of the theses. 

Author Response

Dear reviewer,

thank you very much for your very helpful review.

This article examines the interplay between trinitarian and anti-Pelagian themes in Augustine's Enchiridion.  It is in some ways difficult to assess.  On one side, the author knows the relevant primary and secondary material very well--it seems clear that the article is, to a considerable degree, drawn from work carried out for a book-length project--and is able to make intelligent identifications of some of the theological dynamics that play out in the text.  On the other side, it is not clear that the author has put his or her grasp of the material to work to develop a real original contribution to scholarship.  The commentary is offered almost entirely in a descriptive mode; a fair bit of the commentary--e.g., the noting of overlaps between Ench and other texts and cross-referencing to other scholarship--would be more at home as notes in a critical edition of Ench than in a journal article.  There is something of an 'of course-ness' to the claims that the author demonstrates--i.e., above all, *of course* both anti-Pelagian and trinitarian conceptions would appear and intersect with each other in Enchiridion.  Could we possibly expect Augustine to provide a handbook of Christian faith without drawing from the conceptions that he had spent years developing?

The first point is true. 

I understand the criticism. To some degree this article tries simply to describe how Augustine engaged with his Trinitarian doctrine in the Enchiridion and in his anti-Pelagian oeuvre. As far as I know this descriptive work is still a desideratum, therefore, I did not alter the descriptive chapters decisively. However, I strengthened the notion that the Enchiridion significantly differs from other interpretations of the Creed: I argue that we should understand the argumentation in the Enchiridion as resembling the anti-Pelagian argumentation in which Augustine mostly omits explicit references to his Trinitarian doctrine. Thus, it is maybe not self-evident how Augustine refers to the trinity in the Enchiridion.

Moreover, I tried to integrate the references to anti-Pelagian works better into the argumentation.

In the end, I think the article does enough to merit publication, particular in a special issue of a journal rather than as a standalone original piece.  A few thoughts that might be pursued in a revision:

  1. Structurally, the article may try to do too much even as it risks doing too little.  The risk of doing too little resides in the descriptive 'of course-ness' of the enterprise.  The danger of doing too much resides in the range of topics that the essay treats.  Eight different sections in an essay of ten pages (in the electronic version available to me) is a lot.  Several of them are quite flat descriptions of what is in Ench with fairly workmanlike notes of overlaps and cross-references.  Might fewer sections with more original or critical reflection--sections 7 and 8 are the most obvious candidates for expansion--be richer?
  2. The article takes on different tasks in sections 7 and 8 from the rest of the article.  A couple of thoughts: (i) how far does the question in section 7 fit the logic of Augustine's work?  The idea that a theology can be either trinitarian or Christocentric is distinctively modern; Augustine would not have thought that you could have one without the other, or that the two could possibly compete with each other.  Do we learn anything important by pressing these categories onto Augustine, or would it be more interesting to observe the ways in which he argues exegetically rather than deductively, i.e., he seeks in the first instance to interpret Scripture, and makes deductions from other doctrines primarily when they are needed to resolve ambiguities in the text, rather than supposing, as a matter of principle, that some one doctrine should have primacy, with others conceptions deduced from it.  

(ii) It may be that the best version of this article would be structured around the five theses offered in section 8.  At present, these theses are presented in rather compressed, summary form, and differ in genre from the rest of the essay.  The essay as a whole would, however, be more obviously an original and analytic enterprise if it were structured by these five theses, with the descriptive material that otherwise appears in sections 1-7 reorganised as evidence in support of the theses. 

Good points: I restructured the article so that the four theses (now at the beginning of chapter 3) are congruent to the sections 3 to 6. Hopefully, it  becomes more clear that I outline in the sections 3 to 6 (now 3.1-4) four different modes how Augustine engages his Trinitarian doctrine in the Enchiridion and his anti-Pelagian works.

I have also added some cross-references between the chapter 3 and 4.

Furthermore, I have enhanced the final chapter (now chapter 4). Instead of emphasising the anachronistic antithesis between Christocentric and Trinitarian soteriology I focus more on the main thesis that Augustine increasingly introduced a Trinitarian perspective on his soteriology during the (later) Pelagian controversy. I try to demonstrate that the approach in the Enchiridion is an even more far-reaching defence of the anti-Pelagian soteriology than the Pneumatological soteriology (for example in spir. et litt.). Moreover, I claim that Augustine did not only refer to the “Trinitarian Christology” in ench. and praed. sanct. 1,30-31 due to other discussions (De trinitate, anti-Arian works) but that he also tried to solve several theological problems which evolved in the later Pelagian controversy (doctrine of predestination, God as creator and as saviour).